# RUNX1 maintains the identity of the fetal ovary through an interplay with FOXL2

Barbara Nicol [1], Sara A. Grimm[2], Frédéric Chalmel[3], Estelle Lecluze[3], Maëlle Pannetier[4], Eric Pailhoux[4], Elodie Dupin-De-Beyssat[5], Yann Guiguen [5], Blanche Capel[6] & Humphrey H.-C. Yao [1]*

Sex determination of the gonads begins with fate specification of gonadal supporting cells into either ovarian pre-granulosa cells or testicular Sertoli cells. This fate specification hinges on a balance of transcriptional control. Here we report that expression of the transcription factor RUNX1 is enriched in the fetal ovary in rainbow trout, turtle, mouse, goat, and human. In the mouse, RUNX1 marks the supporting cell lineage and becomes pre-granulosa cell-specific as the gonads differentiate. RUNX1 plays complementary/redundant roles with FOXL2 to maintain fetal granulosa cell identity and combined loss of RUNX1 and FOXL2 results in masculinization of fetal ovaries. At the chromatin level, RUNX1 occupancy overlaps partially with FOXL2 occupancy in the fetal ovary, suggesting that RUNX1 and FOXL2 target common sets of genes. These findings identify RUNX1, with an ovary-biased expression pattern conserved across species, as a regulator in securing the identity of ovarian-supporting cells and the ovary.

[1] Reproductive and Developmental Biology Laboratory, National Institute of Environmental Health Sciences, Research Triangle Park, Durham, North Carolina 27709, USA. [2] Integrative Bioinformatics Support Group, National Institute of Environmental Health Sciences, Research Triangle Park, Durham, North Carolina 27709, USA. [3] Univ Rennes, Inserm, EHESP, Irset (Institut de recherche en santé, environnement et travail) - UMR_S1085, F-35000 Rennes, France. [4] UMR BDR, INRA, ENVA, Université Paris Saclay, 78350 Jouy-en-Josas, France. [5] INRA, UR1037 Fish Physiology and Genomics, F-35000 Rennes, France. [6] Department of Cell Biology, Duke University Medical Center, Durham, North Carolina 27710, USA. *email: humphrey.yao@nih.gov

A critical step that shapes the reproductive identity of the embryo is the sexual differentiation of the bipotential gonads. Supporting cells in the fetal gonads are the first cell population to differentiate and dictate the fate of the gonads. As a consequence, defects in supporting cell differentiation have dire consequences on reproductive outcomes of the individual, from sex reversal to infertility. Supporting cells differentiate into either Sertoli cells, which drive testis development, or pre-granulosa cells, which control ovarian development. It has become clear that supporting cell differentiation and maintenance of their commitment requires a coordinated action of multiple factors that play either complementary, redundant, and even antagonistic roles[1]. For instance, fate decision and maintenance of ovarian identity relies mainly on two conserved elements: the WNT4/RSPO1/β-catenin pathway[2–5] and the transcription factor FOXL2[6–8]. These two elements synergistically promote expression of pro-ovarian genes and, at the same time, antagonize key pro-testis factors such as SOX9 and DMRT1. However, the combined loss of these two key pro-ovarian signaling only results in an incomplete inactivation of ovarian differentiation, suggesting that additional pro-ovarian factors are at play during gonadal differentiation[9,10]. Factors involved in gonad differentiation are generally conserved in vertebrates and even invertebrates, although their position in the hierarchy of the molecular cascade may change during evolution[11]. For instance, the pro-ovarian transcription factor FOXL2 is important for ovarian differentiation/function in human[12], goat[13], and fish[14,15]. The pro-testis transcription factor DMRT1 is highly conserved and critical for testis development in worms, fly[16], fish[17,18], and mammals[19–21].

In this study, we set up to investigate the role of transcription factor RUNX1 in the mouse fetal ovary. In *Drosophila melanogaster*, the *RUNX* ortholog *runt* is essential for ovarian determination[22,23]. In the mouse, *Runx1* mRNA is enriched in the fetal ovary based on transcriptomic analyses[24]. The RUNX family arose early in evolution: members have been identified in metazoans from sponge to human, where they play conserved key roles in developmental processes. In vertebrates, RUNX1 acts as a transcription factor critical for cell lineage specification in multiple organs and particularly in cell populations of epithelial origin[25]. We first characterize the expression profile of *RUNX1* in the fetal gonads in multiple vertebrate species, from fish to human. We then use knockout (KO) mouse models and genomic approaches to determine the function and molecular action of RUNX1 and its interplay with another conserved ovarian regulator, FOXL2, during supporting cell differentiation in the fetal ovary.

## Results

### *Runx1* expression pattern implies a role in ovary development.
The *runt* gene, critical for ovarian determination in the fly[22], has three orthologs in mammals: *RUNX1*, *RUNX2*, and *RUNX3*. Although all three RUNX transcription factors bind the same DNA motif, they are known to have distinct, tissue-specific functions[26]. In the mouse, *Runx1* was the only one with a strong expression in the fetal ovary, whereas *Runx2* and *Runx3* were expressed weakly in the fetal gonads in a non-sexually dimorphic way (Fig. 1a). At the onset of sex determination (Embryonic day 11.5 or E11.5), *Runx1* expression was similar in both fetal XY (testis) and XX (ovary) gonads before becoming ovary-specific after E12.5 (Fig. 1b), consistent with observations by others[24,27]. An ovary-enriched expression of *Runx1* during the window of early gonad differentiation was also observed in other mammals such as human and goat, as well as in species belonging to other classes of vertebrates such as red-eared slider turtle and rainbow

trout (Fig. 1c–f), implying an evolutionarily conserved role of RUNX1 in ovary differentiation.

To identify the cell types that express *Runx1* in the gonads, we examined a reporter mouse model that produces enhanced green fluorescent protein (EGFP) under the control of *Runx1* promoter[28] (Fig. 2 and Supplementary Fig. 1). Consistent with *Runx1* mRNA expression (Fig. 1b), *Runx1*-EGFP was present in both XX and XY gonads at E11.5, then increased in XX gonads and diminished in XY gonads at E12.5 onwards (Fig. 2). At E11.5 in both XX and XY gonads, *Runx1*-EGFP was present in a subset of SF1+/PECAM− somatic cell population, whereas it was absent in the SF1−/PECAM+ germ cells (Fig. 2a–d). In the XY gonads, these *Runx1*-EGFP+ somatic cells corresponded to Sertoli cells, as demonstrated by a complete overlap with SRY, the sex-determining factor that drives Sertoli cell differentiation[29] (Fig. 2e and Supplementary Fig. 1). At this stage, there is no marker for ovarian-supporting cells that allow us to determine which subset of somatic cells were positive for *Runx1*-EGFP in XX gonads. However, at E12.5, when the sex of gonads becomes morphologically distinguishable, *Runx1*-EGFP was specifically detected in the supporting cell lineage of both sexes: strongly in FOXL2+ pre-granulosa cells of XX gonads (Fig. 2g) and weakly in SOX9+ Sertoli cells of XY gonads (Fig. 2f and Supplementary Fig. 1). *Runx1*-EGFP expression was eventually turned off in the fetal testis, while it was maintained in the ovary (Fig. 2h, i). Throughout fetal development of the ovary, *Runx1*-EGFP remained in FOXL2+ pre-granulosa cells (Fig. 3). *Runx1*-EGFP was also detected in the ovarian surface epithelium at E16.5 and birth (arrows in Fig. 3b, c), which gives rise to granulosa cells in the cortex of the ovary[30,31]. *Runx1*-EGFP was also expressed in somatic cells of the cortical region right underneath the surface epithelium at E16.5 and some of these *Runx1*-EGFP+ cells presented a weak expression of FOXL2 (Fig. 3g–i, arrowheads). In summary, *Runx1* marks the supporting cell lineage in the gonads at the onset of sex determination and becomes pre-granulosa cell-specific as gonads differentiate.

### XX *Runx1* KO and *Foxl2* KO share common transcriptomic changes.
Its pre-granulosa cell-specific expression suggests that RUNX1, a factor involved in cell lineage determination[25], could contribute to granulosa cell differentiation and ovarian development. To investigate its role in gonads and avoid early embryonic lethality as a result of global deletion of *Runx1*[32,33], we generated a conditional KO mouse model in which *Runx1* was ablated in the SF1+ gonadal somatic cells[34] (Fig. 4). We characterized the effects of *Runx1* inactivation on XX gonad differentiation at E14.5, a stage where morphological differences between ovary and testis are already established. Although *Runx1* expression was ablated successfully in XX gonads (Fig. 4d), ovarian morphogenesis appeared normal: E14.5 XX *Runx1* KO gonads presented similar size and shape compared with XX control gonads (Supplementary Fig. 2), and differentiation and organization of different gonadal cell populations were similar to control XX gonads (Fig. 4a–c). For instance, XX *Runx1* KO-supporting cells expressed the pre-granulosa cell marker FOXL2 but not Sertoli cell markers SOX9 and AMH (Fig. 4a, b). Supporting cells and germ-cell organization was similar to XX control gonads and did not form Laminin-outlined cord structures typical of testis differentiation (Fig. 4b). Similar to supporting cells, no difference was observed in the COUP-TFII+ interstitium organization between XX *Runx1* KO and XX control gonads (Fig. 4b). Finally, E14.5 XX *Runx1* KO germ cells had initiated meiosis, a typical feature of fetal ovary development, and normal expression of the germ-cell marker *Mvh* was observed (Fig. 4c, d). No significant change was detected for key genes involved in pre-granulosa cell

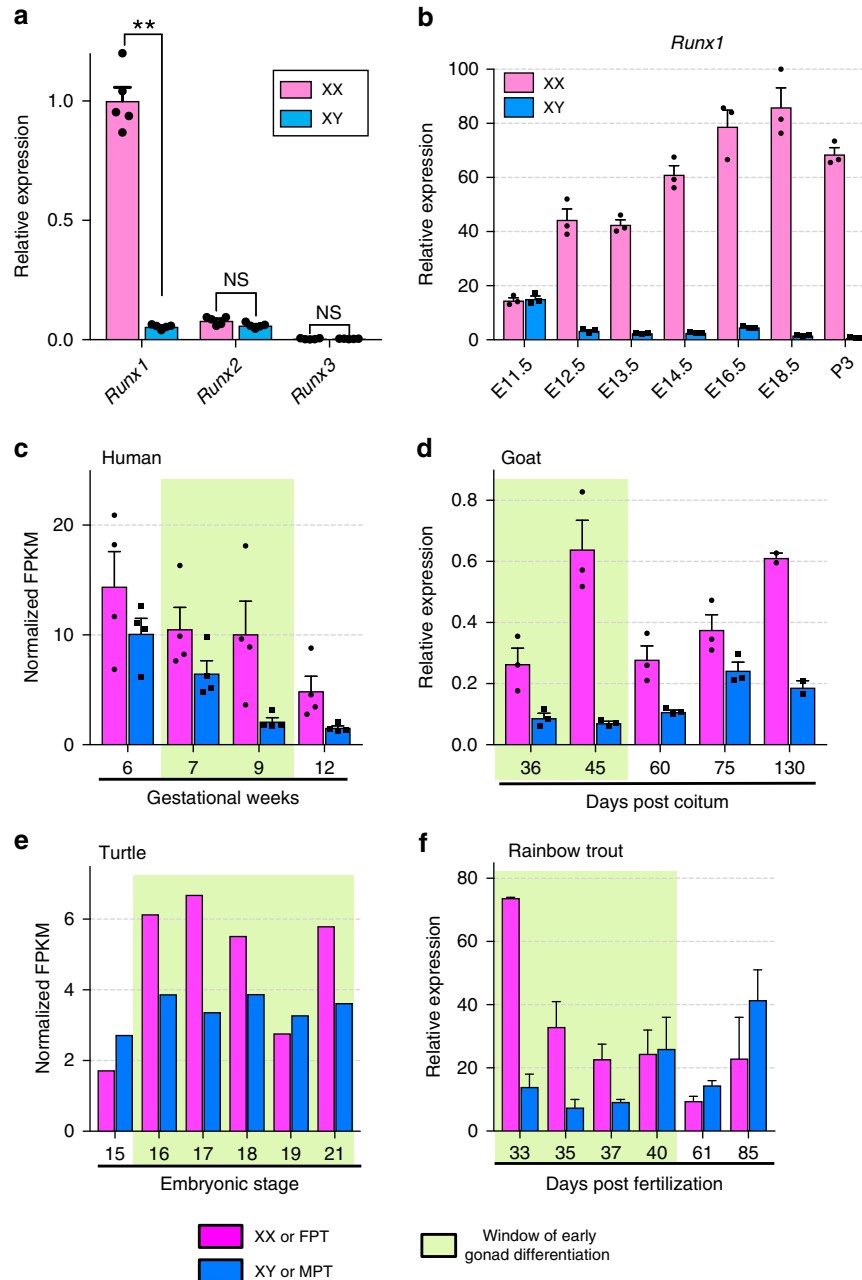

**Fig. 1** RUNX1 expression during gonadal differentiation in various vertebrates. **a** Expression of *Runx1*, *Runx2*, and *Runx3* mRNAs in XX and XY gonads of E14.5 mouse embryos ($n = 5$/sex). Values are presented as mean ± SEM; non-parametric *t*-test, **$p < 0.01$; NS, not significant. **b** Expression time course of *Runx1* mRNA in mouse XX and XY gonads during gonadal differentiation ($n = 3$/stage). Values are presented as mean ± SEM. **c–f** Time course of *RUNX1* mRNA expression in four other vertebrate species, human, goat, red-eared slider turtle, and rainbow trout during gonad differentiation. Values are presented as mean ± SEM. For the turtle, pink and blue bars represent gonads at female-promoting temperature (FPT) of 31 °C and at male-promoting temperature (MPT) of 26 °C, respectively[64]. *RUNX1* expression was analyzed by RNA-seq in human and red-eared slider turtle[64], and by qPCR in goat and rainbow trout. Green highlighted areas represent the window of early gonadal differentiation. Source data are provided as a Source Data file

differentiation (*Foxl2* and *Wnt4*) or Sertoli cell differentiation (*Sox9*, *Fgf9*, *Nr5a1*, and *Amh*; Fig. 4d, e). However, pre-granulosa cell marker *Fst* and fetal Sertoli cell marker desert hedgehog (*Dhh*) were mis-expressed in E14.5 XX *Runx1* KO gonads (Fig. 4d, e).

Similarly, at birth, XX *Runx1* KO gonads maintained a typical ovarian shape, with FOXL2+ pre-granulosa cells scattered throughout the gonad and TRA98+ germ cells located mostly in the cortex (Fig. 4f). Despite their normal ovarian morphology, newborn XX *Runx1* KO gonads exhibited an aberrant transcriptomic profile reminiscent of the transcriptome of newborn XX

*Foxl2* KO gonads, such as downregulation of pro-ovarian genes that are direct targets of FOXL2 (Fig. 4g and Supplementary Data 1). *Foxl2* is involved in ovarian differentiation/maintenance in various vertebrate species. In the mouse, loss of *Foxl2* results in normal ovarian morphogenesis at birth, despite aberrant ovarian transcriptome, and eventually leads to masculinization of the ovary postnatally[6]. We found that 41% of the genes differentially expressed in *Runx1* KO were also misregulated in the absence of *Foxl2* in newborn XX gonads (Fig. 4g and Supplementary Data 1). However, contrary to *Foxl2* KO, loss of *Runx1* in the ovary did not result in postnatal sex reversal and the females *Runx1* KO

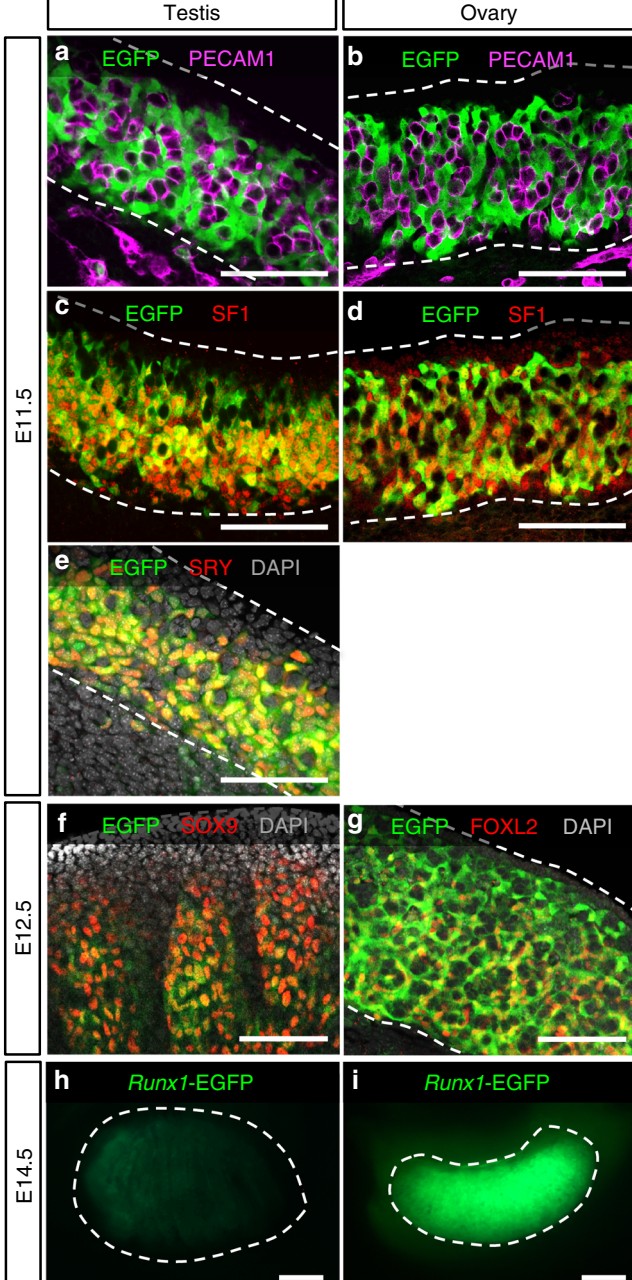

**Fig. 2** *Runx1* is expressed in the supporting cells during gonad differentiation. **a**–**g** Whole-mount immunofluorescence of XY and XX gonads from Tg(Runx1-EGFP) reporter mice at E11.5 and E12.5. Gonads with endogenous EGFP were co-labeled with markers for germ cells/vasculature (PECAM-1; **a**, **b**), somatic cells (SF1; **c**, **d**), Sertoli cells in XY gonads (SRY in **e** and SOX9 in **f**), and for pre-granulosa cells in XX gonads (FOXL2; **g**). Scale bars: 100 μm. Single-channel images are provided for **e**, **f** in Supplementary Fig. 1. **h**, **i** Detection of endogenous EGFP in freshly collected E14.5 gonads. Scale bars: 200 μm. Dotted lines outline the gonads. At least three independent biological replicates were analyzed and the images presented are representative of all replicates

were fertile. One possible explanation for these common transcriptomic changes is that *Runx1* could be part of the same signaling cascade as *Foxl2*. However, analysis of *Runx1* expression in *Foxl2* KO newborn ovaries did not detect any changes in *Runx1* expression in the absence of *Foxl2* (Fig. 4h). Conversely, *Foxl2* expression was not changed in the absence of *Runx1*. These results indicate that *Runx1* and *Foxl2* regulate common sets of genes but they are themselves regulated independently of each other in the fetal ovary.

**Runx1 and Foxl2 double knockout results in partial masculinization of fetal ovaries.** The common transcriptomic changes identified in *Runx1* and *Foxl2* KO newborn ovaries raised the question whether RUNX1 and FOXL2 could play redundant/ complementary roles in supporting cell differentiation. We therefore generated *Runx1/Foxl2* double KO mice (referred as DKO) and compared XX gonads differentiation in the absence of *Runx1*, *Foxl2*, or both (Fig. 5 and Supplementary Figs. 3 and 4). Abnormal development of XX DKO gonads became apparent around E15.5. At this stage, differentiation of supporting cells into Sertoli cells in the testis or pre-granulosa cells in the ovary has been established. For instance, the transcription factor DMRT1, involved in the maintenance of Sertoli cell identity[19], is expressed in Sertoli cells but not pre-granulosa cells (Fig. 5a, e). At E15.5, DMRT1 is also present in a few germ cells in both the testis and ovary[35]. Similar to control ovaries, XX gonads lacking either *Runx1* or *Foxl2* had no DMRT1 proteins in the supporting cells (Fig. 5a–c). However, the combined loss of *Runx1* and *Foxl2* resulted in aberrant expression of DMRT1 in the supporting cells of XX gonads (Fig. 5d). At birth, a time when XX *Foxl2* KO gonads still morphologically resemble ovaries[6,7], XX *Runx1/Foxl2* DKO gonads formed structures similar to fetal testis cords in the center, with DMRT1+ cells surrounding clusters of germ cells (Fig. 5i). Such structure was not observed in XX *Runx1* KO or *Foxl2* KO gonads with the exception that DMRT1 protein started to appear in a few supporting cells in the newborn XX *Foxl2* KO gonads, in what appears to be one of the first signs of postnatal masculinization of *Foxl2* KO ovaries at the protein level (Fig. 5h). Contrary to DMRT1, SOX9 protein, a key driver of Sertoli cell differentiation[36,37], was not detected in XX *Runx1/Foxl2* DKO newborn gonads (Fig. 6). Our results demonstrate that a combined loss of *Runx1/Foxl2* induces partial masculinization of the supporting cells during fetal development of the ovary.

To further characterize the impacts of the combined loss of *Runx1/Foxl2* on ovarian differentiation, we compared the transcriptome of newborn XX *Runx1/Foxl2* DKO gonads with the transcriptomes of XX control, *Runx1*, or *Foxl2* single KO gonads (Fig. 7 and Supplementary Data 2). The heat map for the 918 differentially expressed genes between XX *Runx1/Foxl2* DKO and XX control gonads demonstrated allele-specific impacts: loss of *Runx1* resulted in a mild and often nonsignificant effect on these genes, loss of *Foxl2* had an intermediate/strong effect, and combined loss of *Runx1/Foxl2* yielded the strongest effect (fold change > 1.5; $p < 0.05$ one-way analysis of variance (ANOVA); Fig. 7a and Supplementary Data 3). Gene ontology analysis revealed that the downregulated genes were associated with "ovarian follicle development" and "female gonad development," whereas "male sex determination" was the most significantly enriched process for the upregulated genes (Supplementary Fig. 5). Conforming to the hierarchical clustering (Fig. 7a), comparison of the genes significantly changed in *Runx1/Foxl2* single and DKOs suggested that *Foxl2* was the main contributor to the transcriptional changes observed in *Runx1/Foxl2* DKO (Fig. 7b–g and Supplementary Data 4): 61% of the genes downregulated in DKO were also downregulated in *Foxl2* KO and 43% of the genes upregulated in DKO were also upregulated in *Foxl2* KO. In addition, some genes appeared to be controlled by both *Foxl2* and *Runx1*, and were significantly downregulated or upregulated in all three KOs (Fig. 7b, c). For instance, the genes *Fst* and *Cyp19a*, both involved in granulosa cell differentiation/function[38,39], were downregulated in *Runx1* KO,

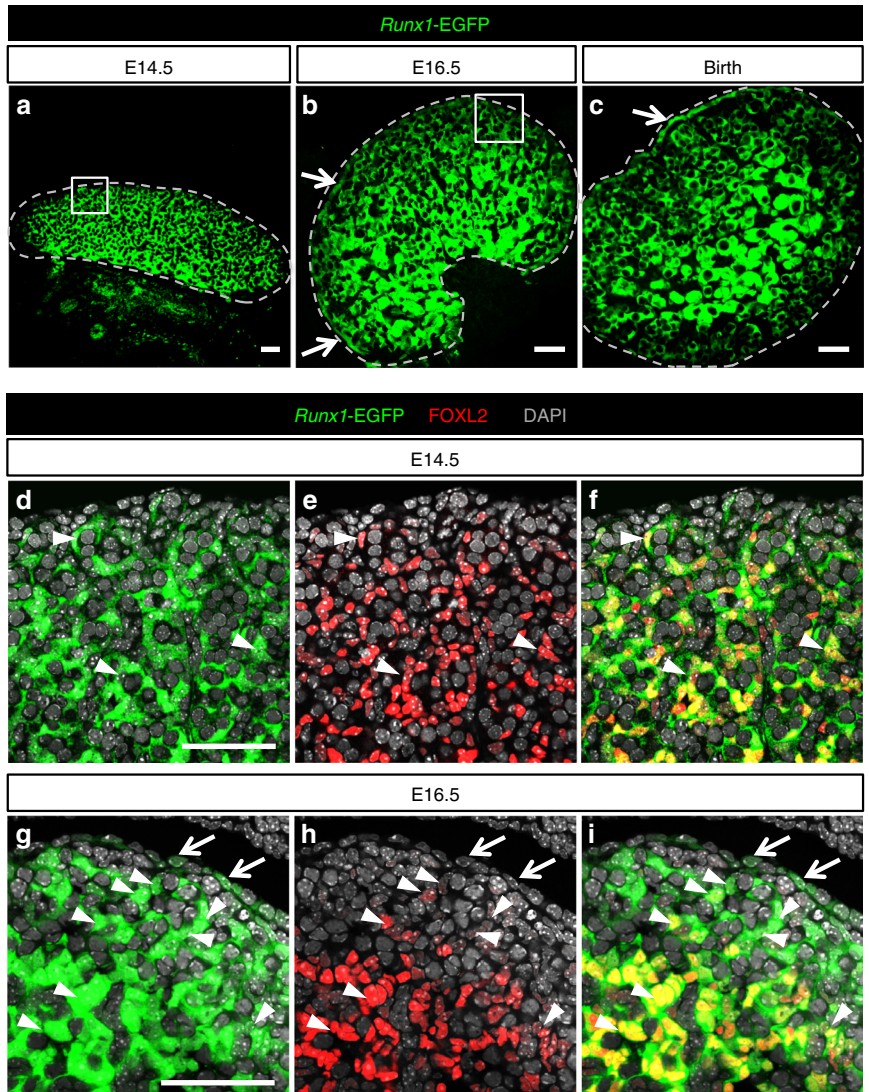

**Fig. 3** *Runx1* expression is maintained in pre-granulosa cells throughout fetal ovarian development. **a–c** Immunofluorescence for EGFP on Tg(*Runx1*-EGFP) ovary sections at E14.5, E16.5, and birth. Scale bars: 50 μm. **d–i** Immunofluorescence for EGFP and the pre-granulosa cell marker FOXL2 at E14.5 (**d–f**) and E16.5 (**g–i**), corresponding to the white square outlined areas in **a** and **b**, respectively. *Runx1* is expressed in pre-granulosa cells throughout ovarian development and the surface epithelium after E14.5. Dotted lines outline the gonads. Arrows: EGFP + ovarian surface epithelium. Arrowheads: EGFP/FOXL2 double-positive cells. Scale bars: 50 μm. At least three independent biological replicates were analyzed and the images presented are representative of all replicates

*Foxl2* KO, and more repressed in *Runx1/Foxl2* DKO (Fig. 7d). On the other hand, *Dhh* was upregulated in all three KOs, with the highest expression in the DKO (Fig. 7f). Finally, some genes were significantly changed in *Runx1/Foxl2* DKO only, suggesting a cumulative effect of *Runx1* and *Foxl2* loss. For instance, *Foxp1*, a gene whose expression is enriched in pre-granulosa cells[40], was significantly downregulated only in *Runx1/Foxl2* DKO (Fig. 7e), whereas *Fgf9*, a Sertoli gene contributing to testis differentiation[41], and *Pdgfc* were significantly upregulated only in *Runx1/Foxl2* DKO (Fig. 7g).

In contrast to XX *Foxl2* single KO gonads (Fig. 5), in which sex reversal only became apparent postnatally[6], XX *Runx1/Foxl2* DKO gonads exhibited masculinization with visible morphological changes before birth. To determine how the additional loss of *Runx1* contributed to the earlier masculinization of *Runx1/Foxl2* DKO ovaries, we identified the genes differentially expressed between XX *Runx1/Foxl2* DKO and XX *Foxl2* KO gonads (Fig. 7h and Supplementary Data 5). Expression of most of these genes was already altered in XX *Foxl2* single KO gonads; however, the

additional loss of *Runx1* exacerbated their mis-expression. For instance, the pro-testis gene *Dmrt1* and *Nr5a1* were significantly upregulated, whereas the pre-granulosa-cell-enriched transcripts *Fst* and *Ryr2* were further downregulated at birth (Fig. 7d, i). On the other hand, the additional loss of *Runx1* did not cause further upregulation of the Sertoli genes *Sox9* and *Amh* at birth, suggesting that *Runx1* does not contribute to their repression in the ovary (Fig. 7j). Overall, the transcriptomic analyses of *Runx1/Foxl2* single and DKOs revealed that *Foxl2* is the main driver of the transcriptomic changes, and that the additional loss of *Runx1* amplifies the mis-expression of genes already altered by the sole loss of *Foxl2*, leading to the failure to maintain pre-granulosa cell identity in the fetal ovary.

**RUNX1 shares genome-wide chromatin occupancy with FOXL2.** The masculinization of XX *Runx1/Foxl2* DKO fetal gonads and the transcriptomic comparisons of newborn XX *Runx1/Foxl2* single and DKO gonads suggest some interplay

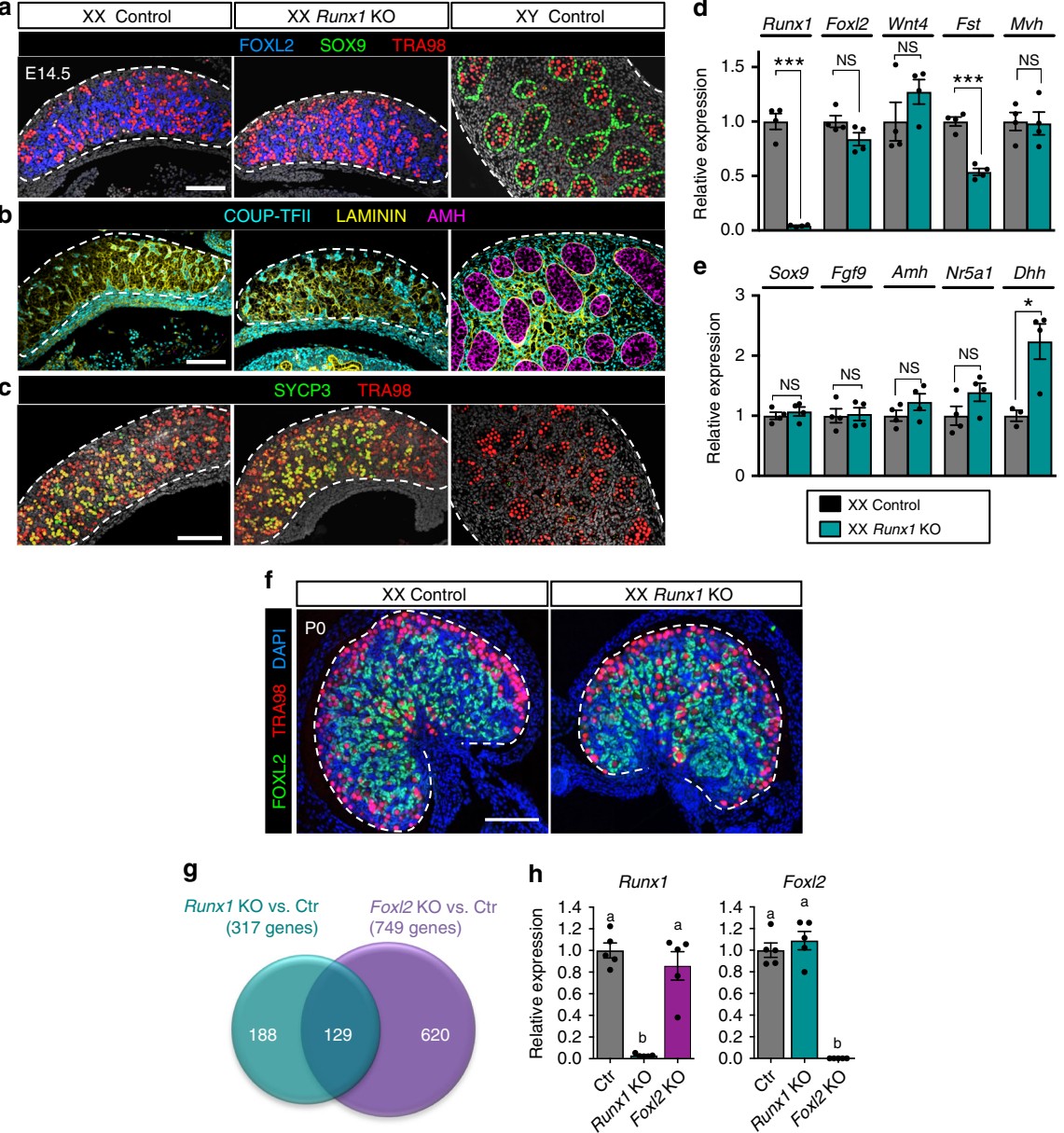

**Fig. 4** Inactivation of *Runx1* in the ovarian somatic cells has minimal impacts on morphogenesis of the ovary with the exception of transcriptomes. Immunofluorescences for **a** pre-granulosa cell marker FOXL2, Sertoli cell marker SOX9, and germ-cell marker TRA98; **b** interstitial cell marker COUP-TFII (encoded by *Nr2f2* gene), Laminin, and Sertoli cell marker AMH; **c** meiotic cell marker SYCP3 and germ-cell marker TRA98 in E14.5 XX control, XX *Runx1* KO, and XY control gonads. The gray color represents DAPI nuclear staining. Single-channel images are provided in Supplementary Fig. 2. Expression of **d** pre-granulosa or germ-cell-specific genes and **e** Sertoli or Leydig cell-specific genes in XX control and XX *Runx1* KO gonads at E14.5 by quantitative PCR ($n = 5$). Values are presented as mean ± SEM.; unpaired Student's *t*-test; *$p < 0.05$, ***$p < 0.001$; NS, not significant. **f** Immunofluorescence for FOXL2, TRA98, and nuclear counterstain DAPI (blue) in XX control and *Runx1* KO gonads at birth (P0). Scale bar: 100 μm. **g** Venn diagram comparing the 317 genes differentially expressed in XX *Runx1* KO vs. XX Control gonads with the 749 genes differentially expressed in XX *Foxl2* KO vs. XX Control gonads at birth. Forty-one percent of the genes differentially expressed in *Runx1* KO (129/317) were also misregulated in the absence of *Foxl2*. Genes differentially expressed were identified by microarray ($n = 4$/genotype; fold change >1.5, one-way ANOVA $p < 0.05$). **h** Quantitative PCR analysis of *Runx1* and *Foxl2* mRNA expression in XX control, *Runx1* KO, and *Foxl2* KO gonads at birth ($n = 5$/genotype). Values are presented as mean ± SEM. One-way ANOVA, $p < 0.05$. Bars with different letters (**a**, **b**) are significantly different. For all experiments, controls are wild-type littermates of *Runx1* KO mice. Source data are provided as a Source Data file. For the immunofluorescences, at least three independent biological replicates were analyzed and the images presented are representative of all replicates

between RUNX1 and FOXL2 to control pre-granulosa cell identity. The fact that RUNX1 and FOXL2 are both transcription factors expressed in the pre-granulosa cells (Fig. 3 and Supplementary Fig. 1) raised the question whether this interplay could occur directly at the chromatin level. We have previously identified FOXL2 chromatin occupancy during ovarian differentiation

by chromatin immunoprecipitation (ChIP) followed by whole-genome sequencing[42] at E14.5, a time where FOXL2 and RUNX1 expression fully overlaps (Fig. 3). We performed additional de novo motif analyses on the genomic regions bound by FOXL2 in the fetal ovary, and discovered that several other DNA motifs were co-enriched with FOXL2 DNA motif (Fig. 8a). RUNX DNA

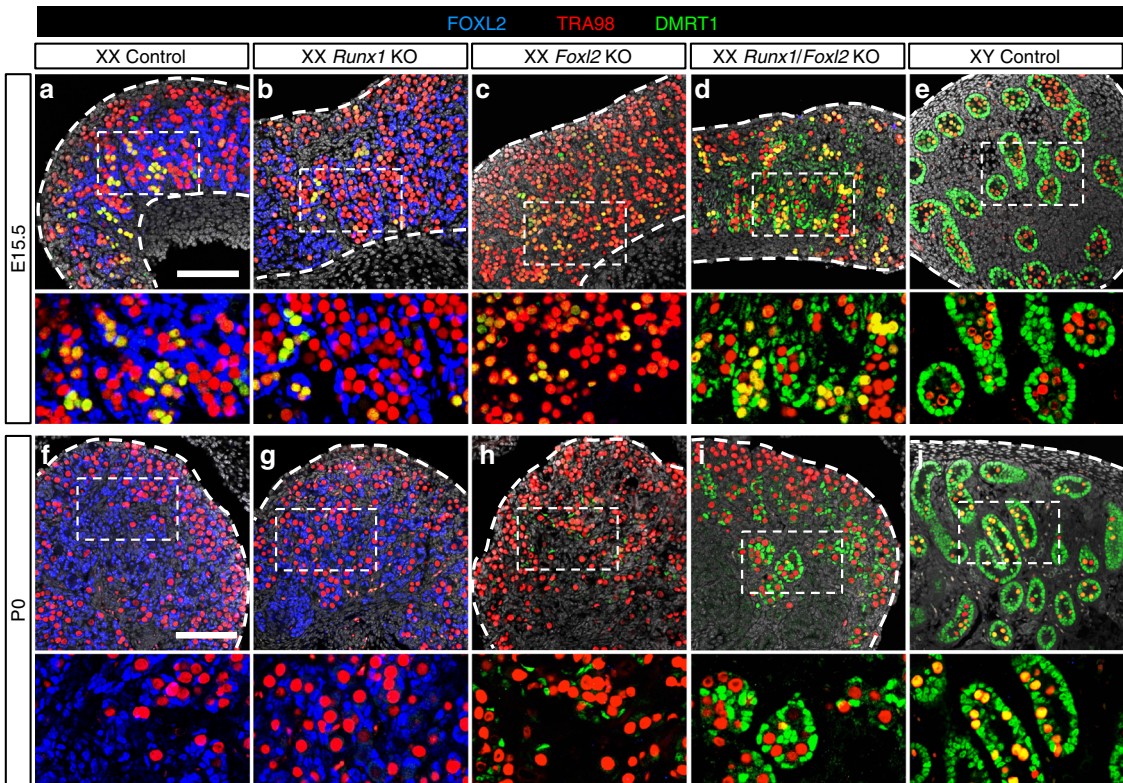

**Fig. 5** Combined loss of *Runx1* and *Foxl2* masculinizes the fetal ovaries. **a–j** Immunofluorescence for Sertoli cell and germ-cell marker DMRT1, germ-cell marker TRA98, and granulosa cell marker FOXL2 in XX control, XX *Runx1* KO, XX *Foxl2* KO, XX *Runx1/Foxl2* double knockout, and XY control gonads at E15.5 (**a–e**) and at birth (**f–j**). Controls correspond to wild-type littermates. The gray represents DAPI nuclear staining. Dotted lines outline the gonads. Higher magnifications are shown for the outlined boxes in **a–e** and **f–j**, respectively. Scale bars: 100 μm. At least three independent biological replicates were analyzed and the images presented are representative of all replicates. Single-channel images are provided in Supplementary Figs. 3 and 4

motif was the second most significantly co-enriched motif. The other motifs were for CTCF, a factor involved in transcriptional regulation, enhancer insulation, and chromatin architecture[43], and for the DNA motif recognized by members of the nuclear receptor family including liver receptor homolog-1 (LRH-1 encoded by *Nr5a2*) and SF1 (encoded by *Nr5a1*), a known co-factor of FOXL2[44,45]. DNA motifs for TEAD transcription factors of the Hippo pathway, ETS, NFYA, and GATA4, another factor involved in gonad differentiation[46], were also significantly enriched. The enrichment of RUNX motif with FOXL2-binding motif suggests that RUNX1, the only RUNX also expressed in pre-granulosa cells, could bind similar genomic regions to FOXL2 in the fetal ovary. To confirm this hypothesis, we performed ChIP-sequencing (ChIP-seq) for RUNX1 in E14.5 ovaries (Supplementary Data 6), the same stage as FOXL2 ChIP-seq[42]. The top de novo motif identified in RUNX1 ChIP-seq ($p < 1e - 559$) matched the RUNX motif[47] (Fig. 8b) and corresponded to the motif that was co-enriched with FOXL2 in FOXL2 ChIP-seq (Fig. 8a). A total of 10,494 RUNX1-binding peaks were identified in the fetal ovary, with the majority of the peaks located either in the gene body (Fig. 8c; 25% exon and 22% intron) or close upstream of the transcription start site (TSS) (30% Promoter: < 1 kb of TSS; 12% Upstream: −10 to −1 kb of TSS). Comparison of genome-wide chromatin binding of RUNX1 and FOXL2 in the fetal ovary revealed significant overlap: 54% (5619/10,494) of RUNX1 peaks overlapped with FOXL2 peaks (Fig. 8d).

The transcriptomic data from *Runx1/Foxl2* DKO ovaries provided us a list of genes significantly changed as a result of the absence of *Runx1*, *Foxl2*, or both (Fig. 7). To identify potential direct target genes of RUNX1 or/and FOXL2, we focused on the

918 genes differentially expressed in *Runx1/Foxl2* DKO ovaries and determined which genes were nearest to RUNX1 or/and FOXL2-binding peaks (Fig. 9a and Supplementary Data 7). More than 50% of these genes (492/918; Fig. 9a) were the closest gene to RUNX1 or/and FOXL2 peaks. Some of these genes were nearest to only FOXL2 peaks (116 genes in Fig. 9a). For example, *Pla2r1*, a transcript enriched in pre-granulosa cells[40] and similarly downregulated in both *Foxl2* KO and *Runx1/Foxl2* DKO fetal ovaries (Fig. 9b), contained two FOXL2-specific peaks, one in the promoter and one in the first intron. On the other hand, 102 genes (Fig. 9a) had RUNX1-specific peaks near their genomic locations. For instance, *Ryr2*, another transcript enriched in pre-granulosa cells[40], was strongly downregulated in both *Runx1* KO and *Runx1/Foxl2* DKO fetal and newborn ovaries (Figs. 7i and 9c), and contained one RUNX1-specific peak in its intronic region. Finally, 274 genes were the closest genes to the peaks for both RUNX1 and FOXL2, with the majority of them (197 genes) nearest to overlapping peaks for RUNX1 and FOXL2 (Fig. 9a). Most of these genes were downregulated in *Runx1/Foxl2* DKO ovaries (Supplementary Data 7). For instance, the pre-granulosa cell-enriched genes *Fst* and *Itpr2*, both downregulated in *Runx1/Foxl2* single and DKO ovaries (Figs. 7d and 9d), contained common binding peaks for FOXL2 and RUNX1 (Fig. 9a, d). For *Fst*, this binding of RUNX1 and FOXL2 was located in its first intron, in the previously identified regulatory region that contributes to its expression[42,48]. On the other hand, Sertoli cell-enriched gene *Dmrt1*, which was strongly upregulated in *Runx1/Foxl2* DKO (Figs. 5 and 7i), contained a common binding site for FOXL2 and RUNX1 near its promoter (Fig. 9a). Taken together, our results reveal that RUNX1, a transcription factor expressed in

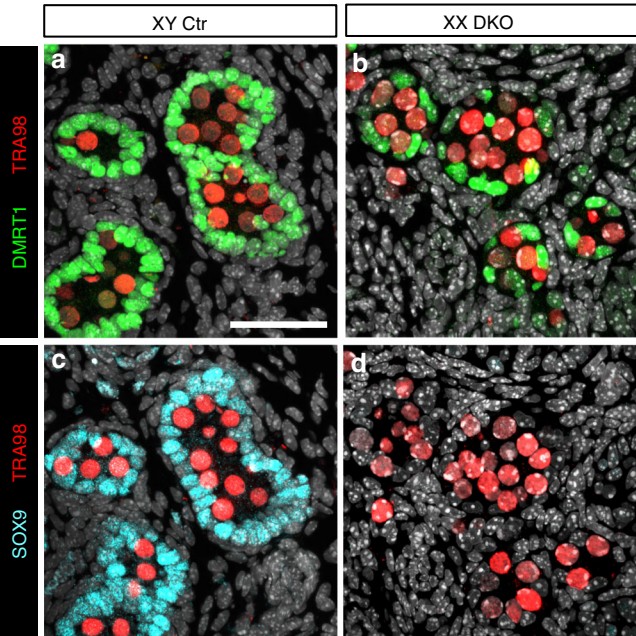

**Fig. 6** SOX9 protein is not detected in *Runx1/Foxl2* DKO newborn XX gonads. Immunofluorescence for Sertoli cell markers DMRT1 and SOX9 and germ-cell marker TRA98 on consecutive sections in XY control (**a**, **c**) and XX *Runx1/Foxl2* DKO gonads (**b**, **d**) at birth. Scale bar: 50 μm. At least three independent biological replicates were analyzed and the images presented are representative of all replicates

the fetal ovary of various vertebrate species, contributes to ovarian differentiation and maintenance of pre-granulosa cell identity through an interplay with FOXL2 that occurs at the chromatin level.

## Discussion

The molecular events that specify pre-granulosa cell fate are complex and nonlinear, involving several signaling pathways that have redundant and complementary functions. This is in contrast to the fetal testis, where the molecular pathway driving its differentiation appears linear and sequential. Removal of one of the top pieces in the testis differentiation pathway has a domino effect that prevents induction of downstream events. This is exemplified by the complete gonadal sex reversal in gain- or loss-of-function mouse models for SRY or SOX9, the two transcription factors responsible for the initiation of the testis morphogenesis[29,36,49]. This is not the case in the mouse ovary, where no single-gene loss/mutation results in a complete ovary-to-testis sex reversal. For instance, defects in the WNT4/RSPO1/β-catenin pathway or loss of *Foxl2* causes a late or postnatal partial ovary-to-testis sex reversal, whereas the combined loss of *Foxl2* and elements of the WNT4/RSPO1/β-catenin pathway (*Wnt4* or *Rspo1*) leads to ovary-to-testis sex reversal more pronounced than each single KO model in the mouse[9,10]. In this study, we demonstrated that *Runx1* contributes to the molecular network controlling pre-granulosa cell differentiation. Loss of *Runx1* in somatic cells of the ovaries altered ovarian transcriptome but did not affect ovarian morphogenesis during fetal development. In contrast, combined loss of *Runx1* and *Foxl2* compromised pre-granulosa cell identity. Loss of *Runx1* or *Foxl2* affected common sets of genes, suggesting some redundancy in their functions in the fetal ovary. These transcriptomic changes were enhanced in the absence of both genes, reaching a threshold that masculinized the fetal ovary. Based on the phenotypic and transcriptomic analyses, we propose that FOXL2 is the dominant

player and RUNX1 acts as a supporting player. For instance, global KO of *Foxl2* in XX gonads results in postnatal sex reversal[6], whereas the XX *Runx1* single KO has no such impacts. In addition, loss of *Foxl2* leads to transcriptomic changes more prominent than loss of *Runx1* in newborn ovaries, even though they affect common sets of genes. However, the additional loss of *Runx1* amplifies the transcriptomic changes caused by the loss of *Foxl2*, resulting in fetal masculinization. One of the most striking changes in *Runx1/Foxl2* DKO ovaries is the expression of DMRT1 in the fetal supporting cells. DMRT1 is a key driver of Sertoli cell differentiation and testis development in various species[16,21]. In the fly, *doublesex* (*dsx*), an ortholog of mammalian *DMRT1*, controls testis differentiation[16]. Intriguingly, *runt*, the fly ortholog of *RUNX1*, tips the balance toward ovarian determination by antagonizing the testis-specific transcriptional regulation of *dsx*[22]. In the mouse, testis differentiation is not controlled by DMRT1 but by SOX transcription factor SRY and its direct target SOX9. However, RUNX1 does not appear to contribute to the repression of the key pro-testis gene *Sox9* in the fetal ovary and SOX9 protein was not detected in *Runx1/Foxl2* DKO ovary at birth. This is in contrast with the phenotype of *Wnt4/Foxl2* DKO newborn ovaries where SOX9 was upregulated, and as a consequence the ovaries were more masculinized[10]. Overall, our findings suggest that slightly different pro-ovarian networks control the repression of the evolutionary conserved pro-testis genes *Sox9* and *Dmrt1*: *Sox9*, which plays a primary role in Sertoli cell differentiation in the mouse, is repressed by an interplay between the WNT4/RSPO1/β-catenin and FOXL2[9,10]. On the other hand, *Dmrt1*, which has taken a secondary role in Sertoli cell differentiation in the mouse, is repressed by an interplay between RUNX1 and FOXL2. The fact that RUNX1 does not appear to control *Sox9* may be the reason why RUNX1 only plays a secondary role in granulosa cell differentiation/maintenance, in contrast to the dominant roles of the WNT pathway and FOXL2. It would be interesting to determine the role of RUNX1 in species for which DMRT1 plays a more prominent role in initiation of differentiation of the testis.

Seeking the mechanisms underlying the interplay between RUNX1 and FOXL2 in the regulation of pre-granulosa cell identity, we identified that RUNX DNA-binding motif is significantly co-enriched with FOXL2 motif in genomic regions bound by FOXL2 in the fetal ovary. The fact that RUNX1 genome-wide chromatin occupancy partially overlaps with FOXL2 in the fetal ovary, that RUNX1 and FOXL2 are expressed in the same cells at the time of the ChIP-seqs, that their loss affects common set of genes, and that the DKO results in gonad masculinization, altogether support the model in which RUNX1 and FOXL2 jointly occupy common chromatin regions that control the maintenance of pre-granulosa cell identity. By themselves, RUNX proteins are weak transcription factors and they require other transcriptional regulators to function as either repressors or activators of transcription[26]. Interplay between RUNX1 and several members of the forkhead transcription factor family has been documented in different tissues. For instance, RUNX1 is a co-activator of FOXO3 in hepatic cells[50]. Similarly, an interplay between RUNX1 and FOXO1/FOXO3 was demonstrated in breast epithelial cells where a subset of FOXO target genes were jointly regulated with RUNX1[51]. Another forkhead protein, FOXP3, acts with RUNX1 to control gene expression in T cells[52] and breast epithelial cells[53]. Such cooperation in various tissues suggest that the interplay between RUNXs and forkhead transcription factors maybe an evolutionary conserved phenomenon. It would be interesting to determine whether the forkhead protein FOXL2 is also able to physically interact with RUNX1 in the fetal ovaries and how this interaction occurs. Our data and single-cell sequencing data[27] demonstrate that RUNX1 starts to

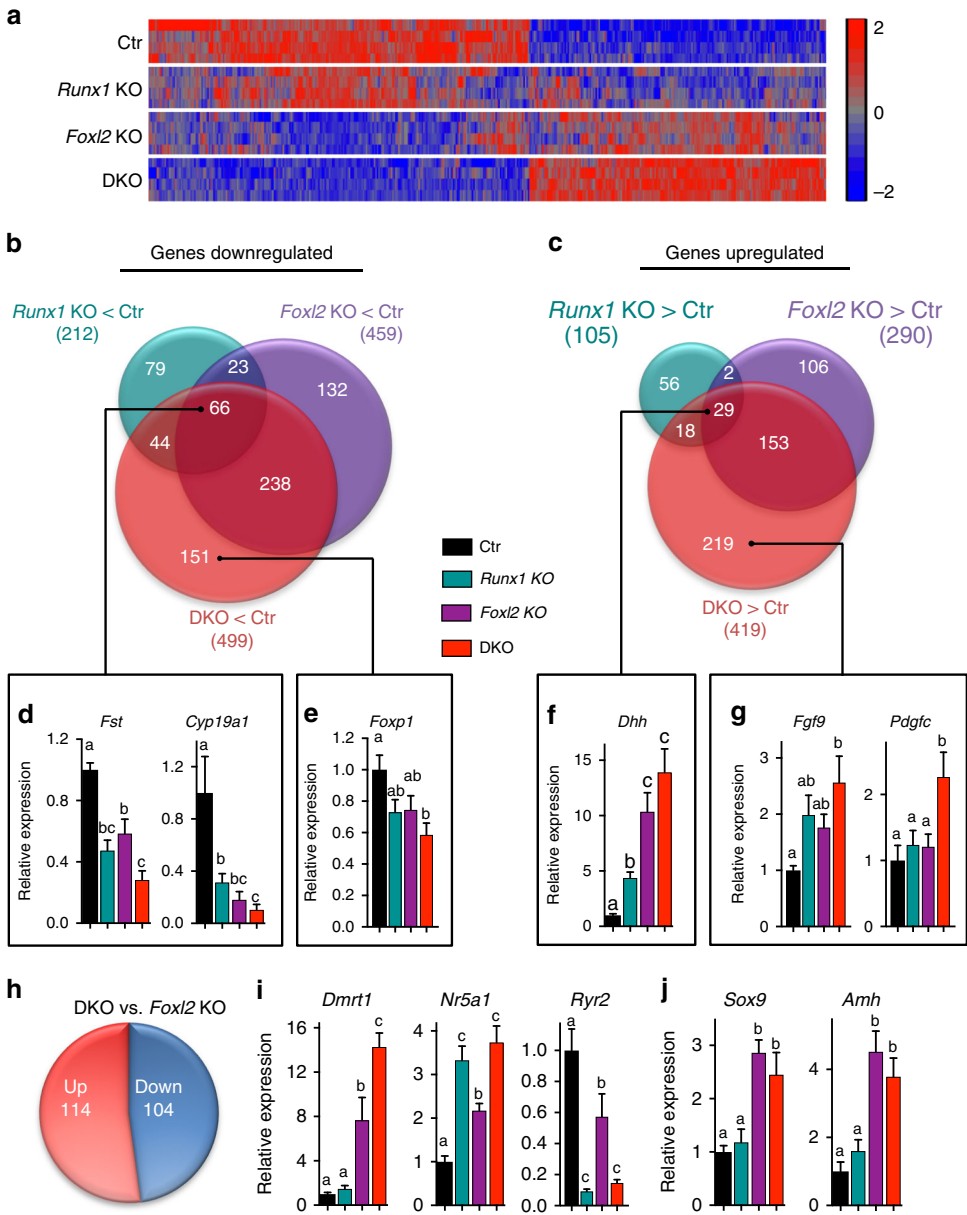

**Fig. 7** Transcriptomic analysis of XX *Runx1/Foxl2* single and double KO ovaries. **a** Heat map for the 918 genes differentially expressed in XX *Runx1/Foxl2* double knockout (DKO) vs. XX control (Ctr) gonads at birth. The heat map shows the expression of these 918 genes in XX control, *Runx1* KO, *Foxl2* KO, and *Runx1/Foxl2* DKO gonads (microarray; $n = 4$/genotype; one-way ANOVA; fold change >1.5, $p < 0.05$). Controls correspond to XX wild-type littermates. **b**, **c** Venn diagram comparing the genes downregulated (**b**) or upregulated (**c**) in XX *Runx1* KO (green circle), XX *Foxl2* KO (purple circle), and XX *Runx1/ Foxl2* DKO (red circle) gonads at birth. **d–g** Validation by quantitative PCR of genes identified in the Venn diagrams as significantly downregulated in all three KO (**d**), or only in *Runx1/Foxl2* DKO (**e**), or significantly upregulated in all in all three KO (**f**) or only in *Runx1/Foxl2* DKO (**g**). **h**, **i** Identification of the genes differentially expressed in XX *Runx1/Foxl2* DKO vs. XX *Foxl2* KO gonads and validation of candidate genes by quantitative PCR. **j** Expression of *Sox9* and *Amh* is not changed in XX *Runx1/Foxl2* DKO compared with XX *Foxl2* KO gonads. For all the qPCR data, values are presented as mean ± SEM ($n = 5$/genotype). One-way ANOVA, $p < 0.05$. Bars with different letters (**a**, **b**, **c**) are significantly different. Source data are provided as a Source Data file

be expressed in the supporting cells earlier than FOXL2. In other tissues, it was demonstrated that RUNX1 can function as a pioneer factor that allows chromatin remodeling and recruitment of other factors to control gene expression and cell fate[54]. Therefore, it is possible that RUNX1 is first recruited to chromatin regions and the presence of RUNX1 facilitates the recruitment of FOXL2 to maintain pre-granulosa cell identity. Finally, in addition to the genes co-regulated by FOXL2 and RUNX1, we identified genes that were specifically mis-expressed in the absence of *Runx1* but not *Foxl2*. Genome-wide analyses of RUNX1 binding in the fetal ovary also identified genomic regions bound by RUNX1 but not

FOXL2. These results suggest that RUNX1 could also contribute to ovarian development or function independently of FOXL2.

RUNX1 contributes to cell-fate determination in various developmental processes such as hematopoiesis and hair follicle development. Depending on its interplay with other signal transduction pathways or co-factors, RUNX1 controls which path the precursor cells take when they are at the crossroad between cell proliferation/renewal and lineage-specific commitment[25]. We discovered that *Runx1* has an ovary-biased expression during gonad differentiation in various vertebrate species, including turtle, rainbow trout, goat, mouse, and human. In mouse

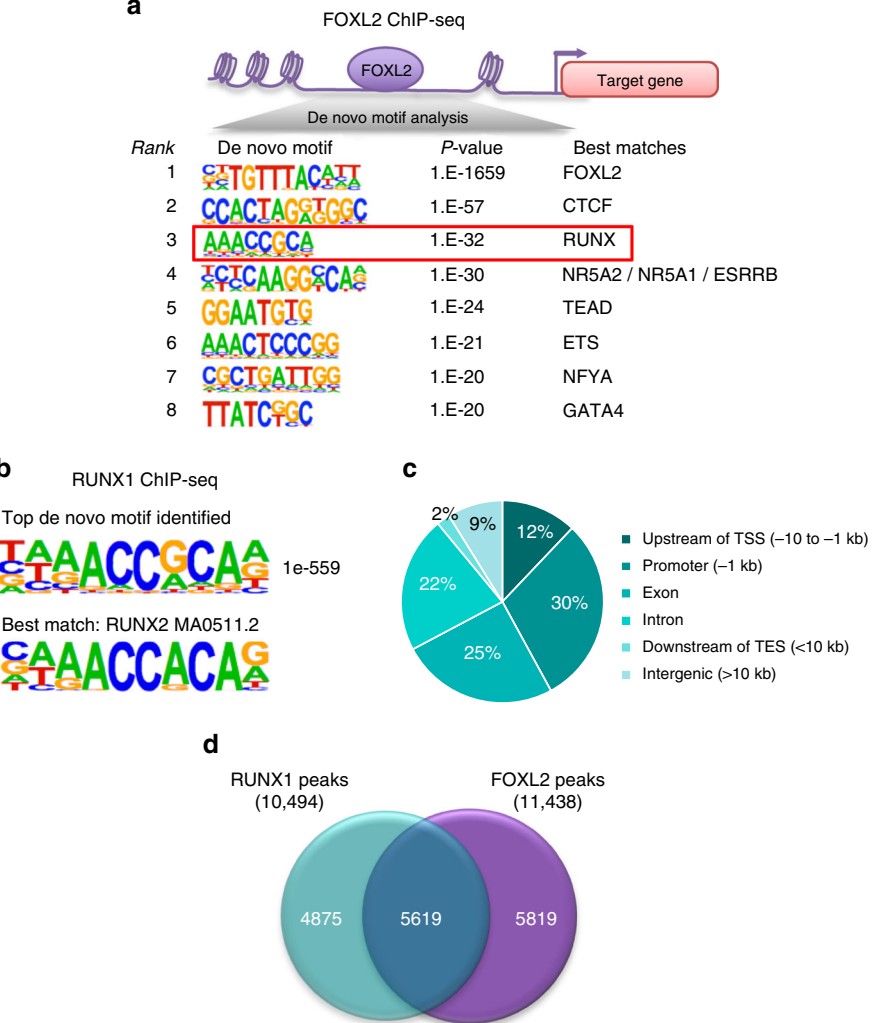

**Fig. 8** RUNX1 and FOXL2 exhibit overlaps in chromatin binding in fetal ovaries. **a** de novo motif analysis of FOXL2 peaks identifies enrichment of RUNX motif along with FOXL2 motif in E14.5 ovaries. **b** The top de novo motif for RUNX1 ChIP-seq in E14.5 ovaries corresponds to a RUNX motif. **c** Distribution of genomic location of the 10,494 RUNX1-binding peaks. TSS, transcription start site; TES, transcription end site. **d** Comparison of RUNX1 (10,494 peaks) and FOXL2 (11,438 peaks) chromatin occupancy in E14.5 ovaries

embryonic gonads, *Runx1* is first detected in the supporting cells in a non-sexually dimorphic way at the onset of sex determination. Although its expression is maintained in the ovary, *Runx1* appears to be actively repressed in the testis between E11.5 and E12.5 as the supporting cells commit to Sertoli cell fate. The suppression of *Runx1* in the fetal testis is corroborated by previously published data from a time-course transcriptomic analysis during early gonad development[55] and single-cell sequencing analysis of SF1+ progenitor cells[27,56]. The time course of Sertoli cell differentiation at the single-cell level revealed that *Runx1* follows a similar spatiotemporal pattern of expression with *Sry*[56]. In the mouse, *Sry* expression in Sertoli cells is quickly turned off after the initiation of testis differentiation and it is suspected that the repression of *Sry* is due to a negative feedback loop by downstream pro-testis genes. The similar pattern of downregulation of *Runx1* in the testis after E11.5 raises the possibility that *Runx1* is downregulated by a similar signaling pathway. Regulation of *Runx1* gene expression is complex and several enhancers that confer tissue-specific expression have been identified[57]. It remains to be determined how *Runx1* expression is controlled in the gonads and how it is actively repressed in the fetal testis.

In contrary to the testes, fetal ovaries maintain expression of *Runx1* in the supporting cells as they differentiate into pregranulosa cells. During ovarian differentiation, granulosa cells arise from two different waves: the first cohort of granulosa cells arises from the bipotential supporting cell precursors that differentiate into either Sertoli cells or pre-granulosa cells during sex determination[58]. The second wave of granulosa cells that eventually populate the cortical region of the ovary appears later in gestation. This second wave arises from LGR5+ cells of the ovarian surface epithelium that ingress into the ovary from E15.5 to postnatal day 4 and eventually become LGR5-/FOXL2+ granulosa cells[30,31]. This timing of establishment of the second cohort of granulosa cells correlates with the expression of *Runx1*-EGFP in a subset of cells in the surface epithelium and in granulosa cells of the ovarian cortex at E16.5 and birth. These results suggest that *Runx1* also marks granulosa cell precursors that will give rise to the second wave of FOXL2+ granulosa cells in the cortex. Therefore, both expression at onset of sex determination and at the surface epithelium/cortex at the time of the second wave of granulosa cells recruitment suggest that *Runx1* is activated in cells that are primed to become supporting/granulosa cells.

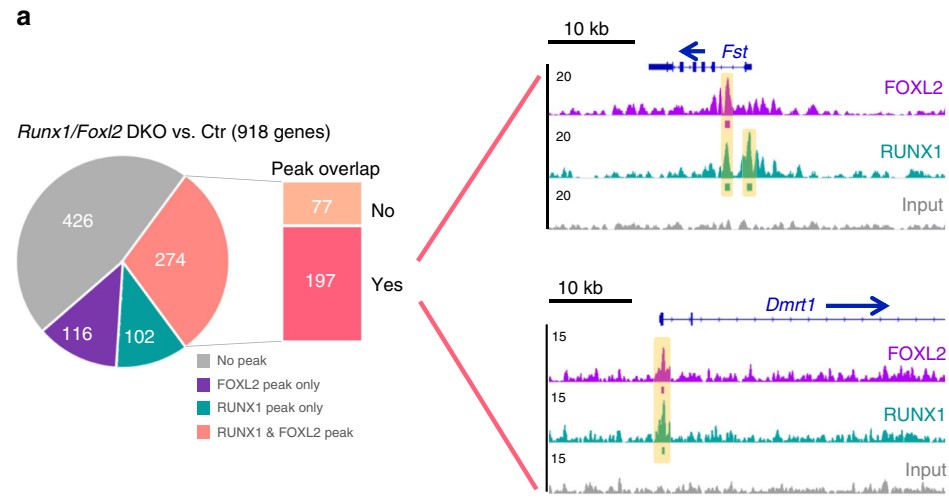

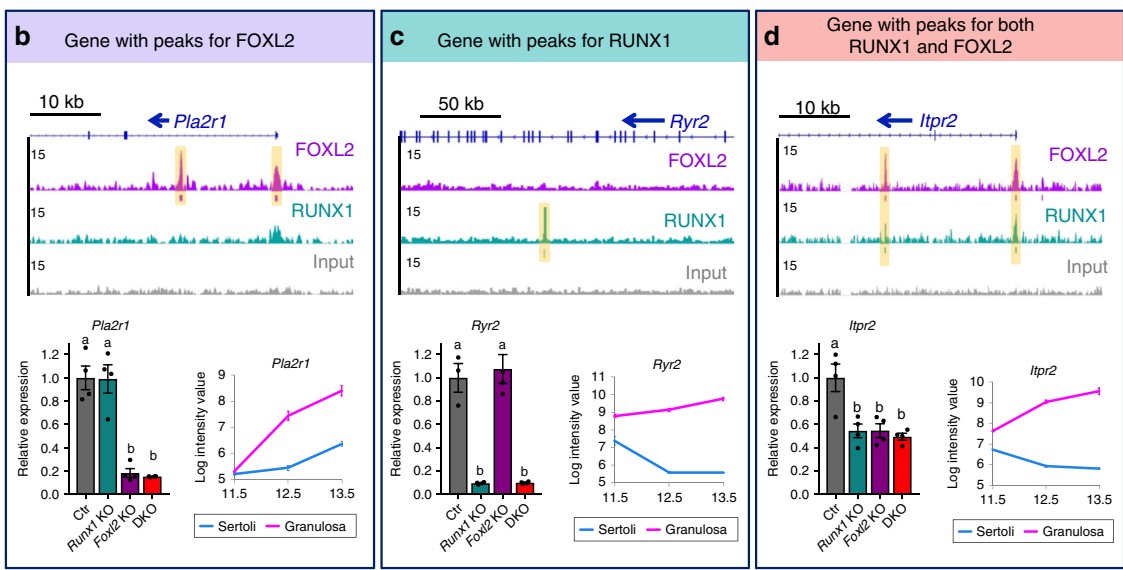

**Fig. 9** Identification of potential direct target genes for RUNX1 and/or FOXL2. **a** Pie-chart identifying the genes significantly changed in XX *Runx1/Foxl2* DKO gonads that are nearest to peaks for FOXL2 and/or RUNX1. Genome browser view of two key genes significantly changed in XX *Runx1/Foxl2* DKO gonads and bound by RUNX1 and FOXL2 in E14.5 ovaries. Blue arrows: gene orientation; orange highlighted area: significant binding peaks identified by HOMER. **b–d** Examples of genes affected in XX *Runx1/Foxl2* DKO gonads and bound by FOXL2 or/and RUNX1. For each gene, we show the genome browser view of RUNX1 and/or FOXL2 binding in E14.5 ovaries, the gene expression by quantitative PCR in XX *Runx1/Foxl2* single and double knockouts gonads at E15.5 (*n* = 4/genotype; mean ± SEM; one-way ANOVA, *p* < 0.05.; bars with different letters (**a**, **b**) are significantly different) and the gene expression in fetal Sertoli and granulosa cells from E11.5 to E13.5[40]. Source data are provided as a Source Data file

Multiple transcription factors often form complex genetic regulatory networks that control cell-fate determination. Genomic sequence motifs or cis-regulatory elements for the supporting cell lineage in the testis were identified by combined analyses of SOX9 and DMRT1 ChIP-seq, and by motif prediction[59]. These "Sertoli cell signatures" are composed of binding motifs for transcription factors critical for Sertoli cell differentiation, including SOX9, GATA4, and DMRT1. These Sertoli cell signatures, present in mammals and other vertebrates, could represent a conserved regulatory code that governs the cascade of Sertoli cell differentiation, regardless of whether it primarily relies on SOX transcription factors such as SRY in mammals or on DMRT1 such as in several vertebrate species. Similarly, one would expect the presence of conserved "granulosa cell signature" genomic regions that confers granulosa cell differentiation. As FOXL2 is a highly conserved gene in granulosa cell

differentiation in vertebrates, we used FOXL2 as an anchor factor to identify other factors that could take part in the regulatory network controlling granulosa cell differentiation/function. Unbiased analyses of the motifs co-enriched with FOXL2 motif in the fetal ovary identified the RUNX motif as one of the most co-enriched motifs. In addition to the RUNX motif, motifs for CTCF, nuclear receptors SF1/LRH-1/ESRRB, and transcription factors TEADs and GATAs were also significantly enriched with FOXL2 consensus motif in FOXL2-bound chromatin regions. For many of these transcription factors, their potential role in gonad differentiation is unknown or limited. For example, the transcription factors of the TEAD family belong to the Hippo pathway, which is involved in the regulation of Sertoli cell gene expression in the fetal gonads[60]. However, the potential involvement of the hippo pathway in granulosa cell differentiation has not been investigated.

In conclusion, we identified RUNX1 as a transcription factor involved in pre-granulosa cell differentiation/maintenance. RUNX1 first delineates the supporting cell lineage and then becomes pre-granulosa cell-specific during gonad development. RUNX1 plays redundant roles with FOXL2 through binding of common chromatin regions and control of common sets of genes to maintain pre-granulosa cell identity in the fetal ovary. Our findings provide insights into the genomic control of granulosa cell differentiation and pave the way for the identification of transcription factors and cis-signatures contributing to the fate determination of granulosa cells and the consequent formation of a functional ovary.

## Methods

**Mouse models.** Tg(*Runx1*-EGFP) reporter mouse was purchased from MMRRC (MMRRC_010771-UCD) and CD-1 mice were purchased from Charles River (stock number 022). *Runx1*[+/−] (B6.129S-*Runx1*[tm1Spe]/J) and *Runx1*[f/f] (B6.129P2-*Runx1*[tm1Tani]/J) mice were purchased from the Jackson Laboratory (stock numbers 005669 and 008772, respectively). *Sf1*-Cre[Tg/Tg] mice[34] (B6D2-Tg(Nr5a1-cre)2Klp) were provided by late Dr. Keith Parker and *Foxl2*[+/−] mice[61] (B6;129-Foxl2 < tm1Gpil > ) were provided by Dr. David Schlessinger (National Institute on Aging), respectively. *Runx1* KO mice (*Sf1Cre*[+/Tg]; *Runx1*[f/−]) were generated by crossing *Runx1*[f/f] females with *Sf1-Cre*[+/Tg]; *Runx1*[+/−] males. Controls were *Sf1-Cre*[+/+]; *Runx1*[+/f] littermates. *Runx1/Foxl2* DKO mice (*Sf1Cre*[+/Tg]; *Runx1*[f/−]; *Foxl2*[−/−]) were generated by crossing *Runx1*[f/f]; *Foxl2*[+/−] females with *Sf1Cre*[+/Tg]; *Runx1*[+/−]; *Foxl2*[+/−] males. This cross also generated the single KOs for *Runx1* (*Sf1Cre*[+/Tg]; *Runx1*[f/−]; *Foxl2*[+/+]) and *Foxl2* (*Sf1Cre*[+/+]; *Runx1*[+/f]; *Foxl2*[−/−]) and control littermates (*Sf1-Cre*[+/+]; *Runx1*[+/f]; *Foxl2*[+/+]). Time-mating was set up by housing female mice with male mice overnight and the females were checked for the presence of vaginal plug the next morning. The day when the vaginal plug was detected was considered embryonic day or E0.5. All experiments were performed on at least four animals for each genotype. All animal procedures were approved by the National Institutes of Health Animals Care and Use Committee, and were performed in accordance with an approved National Institute of Environmental Health Sciences animal study proposal.

**Immunofluorescences.** For the Tg(*Runx1*-EGFP) mice, gonads were collected and fixed in 4% paraformaldehyde for 1–2 h at room temperature. Immunofluorescence experiments were performed on whole gonads at E11.5 and E12.5, and on sections for E14.5, E15.5, E16.5, and P0 (birth) gonads. The EGFP was detected in whole-mount gonads by direct fluorescent imaging and an anti-GFP antibody was used for immunofluorescences on sections. For the different KO models, gonads were fixed in 4% paraformaldehyde overnight at 4 °C and immunofluorescence experiments were performed on paraffin sections of E14.5, E15.5, and P0 gonads as previously described[62]. Briefly, the slides were rehydrated and citrate-based antigen retrieval was performed. The samples (slides or whole-mount samples) were blocked in blocking buffer (5% donkey serum/0.1% Triton X-100 in phosphate-buffered saline (PBS)) for 1 h at room temperature. The samples were then incubated overnight at 4 °C in the primary antibodies diluted in blocking buffer. The next day, the samples were washed three times in 0.1% Triton X-100 in PBS and were incubated for 1 h at room temperature in the secondary antibodies diluted in blocking buffer. The samples were then washed and counterstained with DAPI (4′,6-diamidino-2-phenylindole). The antibodies used in this study are listed in Supplementary Table 1. Whole gonads and sections were imaged under a Leica DMI4000 confocal microscope. For all immunofluorescence experiments, at least three independent biological replicates were analyzed and the image presented in the figures were representative of all replicates.

**Real-time PCR analysis in the mouse.** For the time-course kinetics of *Runx1* expression, fetal gonads from CD-1 embryos at embryonic day E11.5, E12.5, E13.5, E14.5, E16.5, E18.5, and postnatal day P3 were separated from the mesonephros and snap-frozen. For each stage, three biological replicates were collected, with six gonads/replicate for the E11.5 stage and three gonads/replicate for the other stages. For *Runx1* KO analysis, control and KO ovaries were collected at E14.5 (n = 4 biological replicates/genotype). For *Runx1/Foxl2* DKO analysis, control, *Runx1*, and *Foxl2* single and DKO ovaries were collected at E15.5 (n = 4/genotype) and P0 (n = 5/genotype). For all experiments, total RNA was isolated for each replicate using PicoPure RNA isolation kit (Arcturus, Mountain View, CA). RNA quality and concentration were determined using the Nanodrop 2000c and 300–400 ng of RNA was used for cDNA synthesis with the Superscript II cDNA synthesis system (Invitrogen Corp., Carlsbad, CA). Gene expression was analyzed by real-time PCR using Bio-Rad CFX96™ Real-Time PCR Detection system. Gene expression was normalized to *Gapdh*. The Taqman probes and primers used to detect transcript expression are listed in Supplementary Tables 2 and 3. Data were analyzed using Prism GraphPad Software by unpaired Student's *t*-test or by ANOVA $p < 0.05$. Values are presented as mean ± SEM.

**Runx1 expression in other species.** For the rainbow trout, *Runx1* expression during gonadal development was assessed by quantitative PCR[63]. Species-specific primers used are listed in Supplementary Table 3. For the red-eared slider turtle, *Runx1* expression during gonadal development was assessed at Female-Promoting Temperature of 31 °C and at Male-Promoting Temperature of 26 °C by RNA-sequencing (RNA-seq)[64]. For the goat, *Runx1* expression during gonadal development was assessed by quantitative PCR and two to three biological replicates were used for each stage of development[65]. Values are presented as mean ± SD. All goat-handling procedures were conducted in compliance with the guidelines on the Care and Use of Agricultural Animals in Agricultural Research and Teaching in France (Authorization number 91–649 for the Principal Investigator, and national authorizations for all investigators. Approval from the Ethics Committee: 12/045). For the human, *Runx1* expression during gonadal development was assessed by RNA-seq. Human fetuses (6–12 GW) were obtained from legally induced normally progressing terminations of pregnancy performed in Rennes University Hospital in France. Tissues were collected with women's written consent, in accordance with the legal procedure agreed by the National agency for biomedical research (authorization #PFS09-011; Agence de la Biomédecine) and the approval of the Local ethics committee of Rennes Hospital in France (advice # 11–48).

**Microarray analysis.** Gene expression analysis of control, *Runx1* KO, *Foxl2* KO, and *Runx1/Foxl2* DKO ovaries was conducted using Affymetrix Mouse Genome 430 2.0 GeneChip® arrays (Affymetrix, Santa Clara, CA) on four biological replicates (one P0 gonad per replicate) for each genotype. Fifty nanograms of total RNA were amplified and labeled as directed in the WT-Ovation Pico RNA Amplification System and Encore Biotin Module protocols. Amplified biotin-aRNA (4.6 μg) was fragmented and hybridized to each array for 18 h at 45 °C in a rotating hybridization. Array slides were stained with streptavidin/phycoerythrin utilizing a double-antibody staining procedure and then washed for antibody amplification according to the GeneChip Hybridization, Wash and Stain Kit, and user manual following protocol FS450-0004. Arrays were scanned in an Affymetrix Scanner 3000 and data were obtained using the GeneChip® Command Console Software (AGCC; Version 3.2) and Expression Console (Version 1.2). Microarray data have been deposited in GEO under accession code GSE129038. Gene expression analyses were conducted with Partek software (St. Louis, Missouri) using a one-way ANOVA comparing the Robust Multichip Average (RMA)-normalized log2 intensities. A full dataset Excel file containing the normalized log2 intensity of all genes for each genotype and a graphic view of their expression is provided in Supplementary Data 2. In order to identify differentially expressed genes, ANOVA was used to determine whether there was a statistical difference between the means of groups and the gene lists were filtered with $p < 0.05$ and fold-change cutoff of 1.5. The heat map was created comparing the genes that were significantly different between control and *Runx1/Foxl2* DKO ovaries. Venn diagrams were generated in Partek by comparing gene symbols between the lists of genes differentially expressed.

**ChIP-seq assays and analysis.** Ovaries from E14.5 CD-1 embryos were separated from the mesonephros, snap-frozen, and stored at −80 °C. RUNX1 ChIP-seq experiments and analyses in E14.5 ovaries were performed using the same protocol than FOXL2 ChIP-seq[42]. Two independent ChIP-seq experiments were performed by Active Motif, Inc., using 20–30 μg of sheared chromatin from pooled embryonic ovaries (n = 100–120 ovaries/ChIP) and 10 μl of RUNX1 antibody[66] (provided by Drs Yoram Groner and Ditsa Levanon, the Weizmann Institute of Science, Israel). ChIP-seq libraries were sequenced as single-end 75-mers by Illumina NextSeq 500, then filtered to retain only reads with average base quality score > 20. Reads were mapped against the mouse mm10 reference genome using Bowtie v1.2 with parameter "-m 1" to collect only uniquely mapped hits. Duplicate mapped reads were removed using Picard tools MarkDuplicates.jar (v1.110). The number of uniquely mapped non-duplicate reads for each biological replicate was 8,932,674 and 15,036,698. After merging the replicate datasets, binding regions were identified by peak calling using HOMER v.4.9[67] with a false discovery rate < 1e − 5. Called peaks were subsequently re-defined as 300mers centered on the called peak midpoints and filtered for a fourfold enrichment over input and over signal. Genomic distribution of RUNX1-bound regions was determined based on Refseq gene models as downloaded from the UCSC Genome Browser as of 9 August 2017. Enriched motifs were identified using HOMER findMotifsGenome.pl de novo motif analysis with parameter "-size given". For RUNX1 and FOXL2 ChIP-seq comparisons, binding peaks that had at least 1 bp in common were considered overlapping. Peaks were assigned to the nearest gene based on RefSeq. Gene lists were analyzed for enrichment using the online tool EnrichR[68]. The ChIP-seq data are available in the ReproGenomics Viewer (https://rgv.genouest.org)[69,70] and Gene Expression Omnibus (GSE128767; http://www.ncbi.nlm.nih.gov/geo/).

**Reporting summary.** Further information on research design is available in the Nature Research Reporting Summary linked to this article.

## Data availability

The authors declare that all data supporting the findings of this study are available within the article and its Supplementary Information files or from the corresponding author

upon reasonable request. Microarray and ChIP-seq data generated in this study have been deposited in the GEO database under accession codes GSE129038 and GSE128767, respectively. The ChIP-seq data are available in the ReproGenomics Viewer (https://rgv.genouest.org). Raw data underlying all reported mean values in graphs are provided in the Source Data File. All other relevant data supporting the key findings of this study are available in the Supplementary Information files. The source data underlying Figs. 1, 4d-e, 4h, 7d-g, 7i-j, and 9b-d are provided as a Source Data file.

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

## Acknowledgements

We thank late Keith Parker (UT Southwestern Medical Center, USA) for the *Sf1*-Cre mice, David Schlessinger (National Institute on Aging, USA) for the *Foxl2*$^{+/-}$ mice, Yoram Groner and Ditsa Levanon (The Weizmann Institute of Science, Israel) for the RUNX1 antibody for ChIP, Susan Brenner-Morton and late Thomas Jessell (Columbia University, USA) for the RUNX1 antibody for immunofluorescences, David Zarkower (University of Minnesota, USA) for the DMRT1 antibody, Ken Morohashi (Kyushu University, Japan) for the SF1 and SOX9 antibodies, and Dagmar Wilhelm (University of Melbourne, Australia) for the SRY antibody. We are grateful to the NIEHS Molecular Genomics Core, Integrative Bioinformatics Support Group, Comparative Medicine Branch, and Cellular and Molecular Pathology Branch for their services. This work was supported in part by the Intramural Research Program (ES102965) of the NIH, National Institute of Environmental Health Sciences in the U.S. The goat study was supported by Agence Nationale de la Recherche in France (ANR-16-CE14-0020). The human fetal gonads study was supported by l'Institut national de la santé et de la recherche médicale (Inserm), l'Université de Rennes 1 and l'Ecole des hautes études en santé publique (EHESP) in France. The study of red-eared slider turtle was supported by a grant (IOS-1256675) from National Science Foundation in the US.

## Authors contributions

B.N. performed the experiments in the mouse. B.N. and H.H.-C.Y. designed the study, analyzed data, and wrote the paper. S.A.G performed bioinformatic analyses. F.C and E.L. analyzed *RUNX1* expression in human fetal gonads. M.P. and E.P. analyzed *RUNX1* expression in the goat. E.D.-D.-P. and Y.G. analyzed *runx1* expression in rainbow trout. B.C. analyzed *Runx1* expression in the red-eared slider turtle. S.A.G., E.L., F.C., M.P., E.P., E.D.-D.-P., Y.G., B.C. and H.H.-C.Y. edited the paper.

## Competing interests

The authors declare no competing interests.
