## [Peer Review File · Nature Communications]

Reviewers' Comments:

Reviewer #1:

Remarks to the Author:

The manuscript by Nicol et al reports a role for the transcription factor RUNX1 in mouse ovary development, and granulosa cell identity more specifically. This is a nice collaborative piece of work, combining data from a number of labs and sources. Runx1 appears to exhibit ovary-enhanced expression in a number of vertebrate species; and interestingly, the genetic interaction between Runx1 and Foxl2 (is it additive or multiplicative/synergistic?) – in which the DKO results in partial masculinisation of the fetal XX gonad – seems to be an example of an ancient interaction between RUNX and forkhead TFs that even extends to fly sex determination. It is good to be able to expand the genetic “toolkit” of ovary development.

As a geneticist, I am mostly happy with the phenotypic analyses of the various mutants, that clearly add RUNX1 to the ‘list’ of pro-ovary factors, along with WNT/beta-catenin and FOXL2 (and GATA4?). The transcriptomic section of the manuscript was a little less convincing, primarily because the ovary at the stage analysed is heterogeneous at the cellular level. This isn’t even discussed, but should be. Also, reading lists of “x out of y genes” statements isn’t always a great read. I understand that many scientists think these sorts of molecular phenotyping exercises “reveal mechanism” but I remain unconvinced. Perhaps the section could be shortened?

The RUNX1 ChIPseq section is a natural extension of the previous work from this lab on FOXL2. But I should reiterate that at 14.5 dpc the ovary is a complex and heterogeneous collection of cells – perhaps in ways we do not currently understand (are the granulosa cells at this stage clearly a homogeneous population?). Heterogeneity in gonadal cells is being clearly revealed by scRNAseq. So, an important question here is whether the authors can demonstrate that RUNX1 and FOXL2 peaks on a given gene represent joint occupancy of these TFs in the very same cell. The authors should refute (or comment on at the very least) the possibility that single TF occupancy in distinct cells (or cell-types) could lead to the same data.

Overall, this is an important piece of work that sheds new light on ovary development. But like other studies that combine genetics and genomics, the two parts don’t always join together seamlessly. I guess I would like the authors to tell the reader (succinctly) which model of FOXL2-RUNX1 interaction is actually supported by the ChIP data – there’s quite a lot of inference at the moment from Foxl2 data and other model systems (which is fine, but it needs to be declared). Do the authors think that FOXL2 is the dominant player and RUNX1 is a support player?

Finally, a comment on the word “safeguard” in the title and other references to “maintenance” in the text. I am pretty sure that RUNX1 is likely a granulosa cell identity maintenance factor, like FOXL2. But the latter’s role was formally established by deletion of the gene in the adult ovary (Uhlenhaut et al) and other experiments that demonstrated cellular sex reprogramming (as opposed to other possible explanations of the phenotype). Moreover, the fetal Foxl2 KO gonad was exhaustively analysed to establish relative normality. This manuscript does not report the same experiments. The Runx1 KO phenotype is examined at P0, but there are no data from the fetal gonad. So, the authors cannot comment on earlier events in the KO gonad – which may or may not be abnormal. It seems that the authors are saying that Runx1 is a maintenance factor because the fetal XX ovary lacking RUNX1 has granulosa cells at birth and is reminiscent of the Foxl2 KO gonad – but this might be to borrow too much from what we know about FOXL2. It is worth making all this clear so that casual readers aren’t confused by the words “safeguard” or “maintenance” and how it is formally established that Runx1 has this function. (Why was the earlier fetal Runx1 KO gonad not analysed?)

Minor commentss

I think that it is best to refer to the XY and the XX gonad, and to not use ‘testis’ or ‘ovary’ until the

differentiated stage. Even here, when sex reversal is in play, XY and XX is preferable. This point applies to Figs. 1 and 2. Figs 4 and 7 would be better with some reference to chromosomal sex.

The number of independent biological samples examined for each sex/genotype studied should be declared, as should whether the image shown is representative of all of these replicates.

All sections in the main text on transcriptomics should clearly state at which stage the experiment is performed and why e.g. line 168.

Have the authors established that Runx1-GFP is a reliable reporter of endogenous Runx1 expression?

Line 352: there should also be a reference to Stevant et al 2019 (Cell Reports too) in which XY and XX single-cell profiles are compared – in order to establish that suppression of Runx1 is sex-specific.

Line 315: Sox9 in XY gonads is also inhibited by enhanced WNT/b-catenin in the absence of ZNRF3 (Harris et al 2018, PNAS) so may be useful to include this ref here.

The establishment of a granulosa cell signature would be good – but is this lineage sufficiently homogeneous to allow this?

Methods: the genetic background (B6?) of all mutants/lines should be described e.g. Sf1Cre, Foxl2 mutant. We tend not to discuss the best background for studying ovary development phenotypes (cf. B6 for males), but there may be one.

Fig. 4: it is worth saying in the legend what the control is (and in any other legend) in case people only look at the figures and legends.

Reviewer #2:

Remarks to the Author:

The manuscript by Nicol et al, "RUNX1 safeguards the identity of the fetal ovary through an interplay 1 with FOXL2" demonstrates that the transcription factor RUNX1 is upregulated during fetal ovary development, is expressed in pregranulosa cells and while the ovary appears morphologically in somatic cell RUNX1 mutants the transcriptome is altered. In addition, ChIP-seq experiments suggest that RUNX1 and FOXL2 regulate a common set of genes. Runx1/Foxl2 double knockout females have gonads that appear like testes. The manuscript is well written, the experiments carefully performed and the work adds to our understanding of ovary development. Several issues need to be addressed as noted below.

1. The title uses the word interplay which suggests that the two proteins directly interact. This should either be tested or the wording changed. Also, the model of how the two proteins together affect target genes should be explained more clearly in the discussion.
2. Results, Line 75. While the Drosophila runt protein has been shown to have a role in sex determination it has not been shown to be specifically important for ovarian differentiation.
3. The authors report that the Runx1 ovaries appear morphologically normal but was the size measured, were there any differences in cell number?
4. Are the Runx1 mutant females sterile, are there any problems with follicle development?

Reviewers' comments:

We are grateful for the constructive comments from the reviewers, and provide point-by-point responses to the comments below.

Reviewer #1 (Remarks to the Author):

The manuscript by Nicol et al reports a role for the transcription factor RUNX1 in mouse ovary development, and granulosa cell identity more specifically. This is a nice collaborative piece of work, combining data from a number of labs and sources. Runx1 appears to exhibit ovary-enhanced expression in a number of vertebrate species; and interestingly, the genetic interaction between Runx1 and Foxl2 (is it additive or multiplicative/synergistic?) – in which the DKO results in partial masculinisation of the fetal XX gonad – seems to be an example of an ancient interaction between RUNX and forkhead TFs that even extends to fly sex determination. It is good to be able to expand the genetic “toolkit” of ovary development.

As a geneticist, I am mostly happy with the phenotypic analyses of the various mutants, that clearly add RUNX1 to the ‘list’ of pro-ovary factors, along with WNT/beta-catenin and FOXL2 (and GATA4?). The transcriptomic section of the manuscript was a little less convincing, primarily because the ovary at the stage analysed is heterogeneous at the cellular level. This isn’t even discussed, but should be. Also, reading lists of “x out of y genes” statements isn’t always a great read. I understand that many scientists think these sorts of molecular phenotyping exercises “reveal mechanism” but I remain unconvinced. Perhaps the section could be shortened?

Response: We appreciate reviewer’s recognition of our work. Following reviewer’s suggestion, we have now modified and shortened the transcriptomic sections (lines 143-153; 189-231). In regards to the concern on heterogenous nature of the cell populations, while the transcriptomic analysis was done on whole ovaries that contain different cell populations, all the genes validated by qPCR for the different comparisons represent genes that are expressed in a sexually dimorphic manner in the differentiating gonads, and most of these genes are expressed specifically in the supporting cells (granulosa and Sertoli cells).

The RUNX1 ChIPseq section is a natural extension of the previous work from this lab on FOXL2. But I should reiterate that at 14.5 dpc the ovary is a complex and heterogeneous collection of cells – perhaps in ways we do not currently understand (are the granulosa cells at this stage clearly a homogeneous population?). Heterogeneity in gonadal cells is being clearly revealed by scRNAseq. So, an important question here is whether the authors can demonstrate that RUNX1 and FOXL2 peaks on a given gene represent joint occupancy of these TFs in the very same cell. The authors should refute (or comment on at the very least) the possibility that single TF occupancy in distinct cells (or cell-types) could lead to the same data.

Response: The major source of heterogeneity in granulosa cell population arises after E15.5 when the second wave of granulosa cells start to appear in the cortex (see for review Nef, Stevant and Greenfield, 2019). By characterizing the *Runx1*-EGFP reporter line (Figures 2-3), we found that before E16.5, *Runx1*, similar to FOXL2, is specifically expressed in the supporting cells corresponding to the first wave of granulosa cells. We have now confirmed these observations with an antibody recognizing endogenous RUNX1, demonstrating that RUNX1+ and FOXL2+ cells correspond to a single cell population before

E16.5 (Figure S1C). We therefore chose to perform the ChIP-seq experiments at E14.5 when RUNX1 and FOXL2 are expressed in the same cells that correspond to the first wave of granulosa cells (as shown in Figure 3 d-f). Our data are also in agreement with recent single cell sequencing in female gonads (Stevant et al., 2019) that shows that at E13.5 (there was no E14.5 time-point) *Runx1* and *Foxl2* are expressed in the same cluster of cells. Therefore, at the stage we performed the ChIP-seq experiments, we are confident that the chromatin binding identified for RUNX1 and FOXL2 correspond to a unique cell population.

Based on the facts that RUNX1 and FOXL2 are expressed in the same cells at the time of the ChIP-seq, that their loss affects common set of genes, that they bind common regions in the chromatin of fetal ovaries, and that RUNX1 DNA binding motif is co-enriched with FOXL2 motif in regions bound by FOXL2, we propose that they may bind these regions at the same time and physically interact in the granulosa cells. This hypothesis is also strengthened by findings from other labs that RUNX1 interact physically with other members of the forkhead family as mentioned in the discussion. Unfortunately, we were not able to confirm the physical interaction between RUNX1 and FOXL2 in the fetal ovaries due to the limited amount of tissue available to perform co-IP and to a lack of reliable RUNX1 antibody for Western Blot in fetal ovaries. We have now included these discussion on FOXL2/RUNX1 interplay in lines 241; 349-354; 364-372.

Overall, this is an important piece of work that sheds new light on ovary development. But like other studies that combine genetics and genomics, the two parts don't always join together seamlessly. I guess I would like the authors to tell the reader (succinctly) which model of FOXL2-RUNX1 interaction is actually supported by the ChIP data – there's quite a lot of inference at the moment from *Foxl2* data and other model systems (which is fine, but it needs to be declared). Do the authors think that FOXL2 is the dominant player and RUNX1 is a support player?

Response: Yes, the reviewer's understanding is correct. Based on the phenotypic and transcriptomic analyses of the single and double KO, we propose that FOXL2 is the dominant player and RUNX1 acts as a supporting player. For instance, global KO of *Foxl2* in XX gonads results in postnatal sex reversal (Ottolenghi et al., 2005) whereas the XX *Runx1* single knockout has no such impacts. In addition, loss of *Foxl2* results in transcriptomic changes stronger than loss of *Runx1* in newborn ovaries, even though they affect common sets of genes. This information is provided lines 191; 198; 228; 317 and 340-345. As indicated in the previous comment, we have now provided additional discussion and clarification on the model of interaction between FOXL2 and RUNX1 during ovarian differentiation based on our mouse models, transcriptomic and ChIP-seq data lines 349-354; 364-372.

Finally, a comment on the word "safeguard" in the title and other references to "maintenance" in the text. I am pretty sure that RUNX1 is likely a granulosa cell identity maintenance factor, like FOXL2. But the latter's role was formally established by deletion of the gene in the adult ovary (Uhlenhaut et al) and other experiments that demonstrated cellular sex reprogramming (as opposed to other possible explanations of the phenotype). Moreover, the fetal *Foxl2* KO gonad was exhaustively analysed to establish relative normality. This manuscript does not report the same experiments. The *Runx1* KO phenotype is examined at P0, but there are no data from the fetal gonad. So, the authors cannot comment on earlier events in the KO gonad – which may or may not be abnormal. It seems that the authors are saying that *Runx1* is a maintenance factor because the fetal XX ovary lacking RUNX1 has granulosa cells at birth and is reminiscent of the *Foxl2* KO gonad – but this might be to borrow too much from what we know about FOXL2. It is worth making all this clear so that casual readers aren't confused by the words "safeguard" or "maintenance" and how it is formally established that *Runx1* has this

function. (Why was the earlier fetal Runx1 KO gonad not analysed?)

Response: First, our study focused specifically on fetal ovary and therefore the term of “safeguard” in the title refer to this particular developmental stage. For reviewer’s information, we also examined the role of RUNX1 in the adult ovary in another study. Contrary to the adult *Foxl2* knockout model, deletion of *Runx1* in adult granulosa cells does not result in sex reprogramming. Second, To further support our claim that fetal ovaries lacking *Runx1* appear normal, we have now added bright field images of the gonads, immunofluorescences for multiple markers of different cell populations and gene expression data for E14.5 XX *Runx1* KO (Figures 4a-e and S2, lines 121-139). These data demonstrate that XX *Runx1* KO gonads are similar to control XX gonads in their size, morphological organization of the different cell populations, and protein expression despite minor transcriptomic changes. We also modified the text lines 142-153 to make our points more clear.

Minor comments

I think that it is best to refer to the XY and the XX gonad, and to not use ‘testis’ or ‘ovary’ until the differentiated stage. Even here, when sex reversal is in play, XY and XX is preferable. This point applies to Figs. 1 and 2. Figs 4 and 7 would be better with some reference to chromosomal sex.

Response: We made the changes in all the figures that the reviewer suggested with the exception of the part regarding other vertebrate species in Fig.1 as not all species presented rely on the same sex determination mechanism. When the comparison among species was made, we kept the terms ovary and testis.

The number of independent biological samples examined for each sex/genotype studied should be declared, as should whether the image shown is representative of all of these replicates.

Response: We have now added this information in the material and methods. For all immunofluorescence experiments, at least 3 biological replicates were used and the images used in the figures are representatives of all replicates.

All sections in the main text on transcriptomics should clearly state at which stage the experiment is performed and why e.g. line 168.

Response: We have now added this information in the results lines 121-123; 167 and 176.

Have the authors established that Runx1-GFP is a reliable reporter of endogenous Runx1 expression?

Response: Yes. We have now found an antibody that recognizes RUNX1 in the gonads and have confirmed that *Runx1*-EGFP accurately reports endogenous RUNX1 expression in the fetal gonads. We also performed co-immunofluorescences with FOXL2 to confirm that indeed RUNX1 and FOXL2 are expressed in the nuclei of the same cells in the fetal ovary. These new data are now presented in Figure S1 and a high magnification is presented below.

Line 352: there should also be a reference to Stevant et al 2019 (Cell Reports too) in which XY and XX single-cell profiles are compared – in order to establish that suppression of Runx1 is sex-specific.
 Line 315: Sox9 in XY gonads is also inhibited by enhanced WNT/b-catenin in the absence of ZNRF3 (Harris et al 2018, PNAS) so may be useful to include this ref here.

Response: We have now added these references in the manuscript.

The establishment of a granulosa cell signature would be good – but is this lineage sufficiently homogeneous to allow this?

Response: We think that the stage we performed the ChIP-seq would allow to identify some potential granulosa cell signatures, corresponding to the differentiation of the first wave of granulosa cells. What could be interesting is to have different time-points of ovarian development to identify some signatures that are specific of early differentiation (first wave then second wave of granulosa cells differentiation) or specific of granulosa cell function and folliculogenesis, through ChIP-seqs for different chromatin marks as well as for several transcription factors that may be involved in granulosa cell differentiation or function.

Methods: the genetic background (B6?) of all mutants/lines should be described e.g. Sf1Cre, Foxl2 mutant. We tend not to discuss the best background for studying ovary development phenotypes (cf. B6 for males), but there may be one.

Response: We have now added the information about the genetic background for all the mouse lines used in this study in the material and methods lines 459-460.

Fig. 4: it is worth saying in the legend what the control is (and in any other legend) in case people only look at the figures and legends.

Response: We have now added this information in the figure legends. All controls are wild-type littermates.

Reviewer #2 (Remarks to the Author):

The manuscript by Nicol et al, “RUNX1 safeguards the identity of the fetal ovary through an interplay 1 with FOXL2” demonstrates that the transcription factor RUNX1 is upregulated during fetal ovary development, is expressed in pregranulosa cells and while the ovary appears morphologically in somatic

cell RUNX1 mutants the transcriptome is altered. In addition, ChIP-seq experiments suggest that RUNX1 and FOXL2 regulate a common set of genes. Runx1/Foxl2 double knockout females have gonads that appear like testes. The manuscript is well written, the experiments carefully performed and the work adds to our understanding of ovary development. Several issues need to be addressed as noted below.

1. The title uses the word interplay which suggests that the two proteins directly interact. This should either be tested or the wording changed. Also, the model of how the two proteins together affect target genes should be explained more clearly in the discussion.

Response: First, we understand reviewer's concern on the word "interplay". However, interplay does not necessarily imply physical interaction. It can be an interaction between two signaling pathway that eventually lean toward common effects. We would appreciate reviewer's suggestion on a better replacement. Second, based on the facts that RUNX1 and FOXL2 are expressed in the same cells at the time of the ChIP-seq, their loss affects common set of genes, they bind common regions in the chromatin of fetal ovaries, and RUNX1 binding motif is co-enriched with FOXL2 motif in regions bound by FOXL2, we propose that they could bind these regions at the same time and physically interact in the granulosa cells. This proposal is also strengthened by findings from other labs that RUNX1 interact physically with other members of the forkhead family as mentioned in the discussion. Unfortunately, we were not able to confirm physical interaction between RUNX1 and FOXL2 in the fetal ovaries due to the limited amount of tissue available to perform co-IP and to a lack of reliable antibodies for Western Blot in fetal ovaries. We have now provided in the manuscript more clear explanations of the potential model of how FOXL2/RUNX1 together affect target genes lines 198-200; 228-231; 317-323; 340-345; 349-354; 364-372; 444-449.

2. Results, Line 75. While the Drosophila runt protein has been shown to have a role in sex determination it has not been shown to be specifically important for ovarian differentiation.

Response: We have now revised the text lines 63 and 328.

3. The authors report that the Runx1 ovaries appear morphologically normal but was the size measured, were there any differences in cell number?

Response: No difference was observed in gonadal size between XX *Runx1* KO and XX control gonads during fetal development (n=4 /genotype). We have now added bright field images of the whole gonads at E14.5 in Figure S2 as well as immunofluorescences for different cell population markers (Figure 4a-c, lines 121-139 and Figure S2) to demonstrate that the size and cellular composition of the XX *Runx1* KO gonad were morphologically similar to that of XX control gonads. We also determined that the germ cell number was not significantly different than in XX control during fetal development based on unchanged *Mvh* gene expression at E14.5 (now provided in Figure 4) and based on cell counting (this result will be included in a separate publication focusing on *Runx1* function in folliculogenesis).

4. Are the Runx1 mutant females sterile, are there any problems with follicle development?

Response: These *Runx1* mutant females are fertile and we have now added this information in the manuscript line 152. Loss of *Runx1* in the granulosa cells, however, results in some defects during follicle development that we are currently investigating and these studies will be the subject of a separate publication.

REVIEWERS' COMMENTS:

Reviewer #1 (Remarks to the Author):

The revised manuscript is an improvement - it is clearer and has a better discussion of the significance of the data.

--

Reviewer #2 (Remarks to the Author):

All of my concerns and the concerns of the other reviewer have been thoroughly addressed.